# The Impact of Big Data Adoption on SMEs' Performance

Mahdi Nasrollahi [1],[*] , Javaneh Ramezani [2],[3] and Mahmoud Sadraei [4]

1    Faculty of Social Sciences, Imam Khomeini International University (IKIU), Qazvin 34149-16818, Iran
2    Faculty of Sciences and Technology, Campus da Caparica, NOVA University of Lisbon,
     2829-516 Caparica, Portugal; m.ramezani@campus.fct.unl.pt
3    Center of Technology and Systems, UNINOVA-CTS, 2829-516 Caparica, Portugal
4    IVE Group, Sydney 2000, Australia; mahmoud.sadraei@ivegroup.com.au
*    Correspondence: m.nasrollahi@soc.ikiu.ac.ir

**Abstract:** The notion of Industry 4.0 encompasses the adoption of new information technologies that enable an enormous amount of information to be digitally collected, analyzed, and exploited in organizations to make better decisions. Therefore, finding how organizations can adopt big data (BD) components to improve their performance becomes a relevant research area. This issue is becoming more pertinent for small and medium enterprises (SMEs), especially in developing countries that encounter limited resources and infrastructures. Due to the lack of empirical studies related to big data adoption (BDA) and BD's business value, especially in SMEs, this study investigates the impact of BDA on SMEs' performance by obtaining the required data from experts. The quantitative investigation followed a mixed approach, including survey data from 224 managers from Iranian SMEs, and a structural equation modeling (SEM) methodology for the data analysis. Results showed that 12 factors affected the BDA in SMEs. BDA can affect both operational performance and economic performance. There has been no support for the influence of BDA and economic performance on social performance. Finally, the study implications and findings are discussed alongside future research suggestions, as well as some limitations and unanswered questions.

**Keywords:** Industry 4.0; big data adoption; SMEs; organizational performance; social performance; structural equation modeling

## 1. Introduction

In recent years, industries have shifted towards the age of Industry 4.0 or digital transformation. This transformation results from contributions from various engineering and manufacturing disciplines and merging technologies, including artificial intelligence and machine learning, neurotechnologies, mobile and cloud computing, sensing, and other "exponential technologies". In the era of Industry 4.0, growing exponentially, data significantly impact obtaining new knowledge, fostering business innovation, and achieving competitive advantage. Accordingly, the utilization of BD with distinct capabilities, features, and specifications has grown rapidly in industrial contexts because of engineering developments and, more specifically, in computer science and information technology. BD with developing data mining techniques supports the so-called smartness of everything everywhere by facilitating real-time, dynamic, self-adaptive, and precise control capabilities [1–3].

Theoretically, there is a wide-ranging discussion about the impact of information technology (IT) on the organization's performance. In such a trend, however, IT contributes to firms' competitiveness, but its role is still relatively unclear [4]. In conditions where information flows in the internet and virtual networks are established with high speed and intensity, the level of information of individuals as the main audiences of this data flow is affected by this trend. Thus, the emergence of new ideas and creativity from each person and staff [5] and, consequently, the necessary creation of a new structure for the

current organizations in the global society, is inevitable and worthy of serious review and contemplation.

In addition to the hardware available in each job system that requires updating in line with the current turbulent era, the software related to each job also needs to be reviewed and updated. On the other hand, knowledge management cycle (KMC) models become essential issues for organizations in this volatile and globalized world in order to make valuable decisions and achieve business outcomes. KMC is a strategic tool for fostering the symbiosis of implicit and explicit knowledge and creating a proper perception of it in organizations [1].

BD is one of the topics that has been considered by managers and decision makers nowadays due to the high speed and volume of data exchange to develop productivity in industries [6].

Since the competitive market becomes more complicated day by day, organizations' decision makers increasingly rely on BD to enhance the visibility of firm operations and organizational performance, predict market trends, facilitate innovation, seize new opportunities, and improve their competitive position [7,8].

Hence, it is necessary to ensure the quality and reliability of the used data. Lack of adequate and reliable data may lead to inappropriate decisions, missing opportunities, consequently financial losses, and damage to organizational reputation [9].

Recently, although BD has become an important issue in different sectors, and numerous studies have been conducted on various methods of data analysis and storage, particularly in the field of industry; still, in small and medium enterprises (SMEs), less attention has been paid. It is a relatively unknown issue for these organizations' managers [10,11].

As reported, adoption of BD is dependent upon technological, organizational, managerial, and environmental factors and positively influences firm performance [12,13]. SMEs, in comparison to large firms, are more flexible and adaptable to change and new technologies. Additionally, being under pressure to compete, lacking financial, required resources, and talent capabilities are significant characteristics that could affect an SME manager's decision making in adopting BD. As such, studies have shown that technology and innovation are strategic priorities for SMEs' growth, and among them, BD will be one of the main drivers [11,14].

BDA enables organizations to create a clear picture of customers and their demands to make well-informed decisions in designing marketing strategies [15]. SMEs, in turn, also require BD to analyze the market and predict customer behavior. BD in SMEs can lead to increased flexibility, efficiency, responsiveness, and the ability to anticipate and meet customer needs, thus providing organizations with a competitive edge [14].

Despite much attention from scholars to BD, the available BDA and firm performance studies are still few and fragmented, especially in SMEs. Notably, most reports to date explaining how performance gain can be achieved come from large and well-established firms, especially in developed countries, and lack in attempts to investigate the main factors that affect SMEs' intention to adopt BD in developing countries such as Iran [16,17].

However, the impact of BDA could vary depending on the type of technology, the size of the organization, and the country under study [12,18,19]. Therefore, there is a need for more in-depth studies for novel approaches and methodologies to gain a broad overview of the factors that impact firm performance and their relationship with organizational intention to adopt BD.

Hence, the present study is designed to evaluate the impact of BDA on SMEs' performance in Iran, an issue that has not been studied yet. In doing so, grounded on previous studies, we tried to identify the key factors that contribute to performance gain and investigate how organizations should approach their BDA strategy to derive business value from BD investments. Therefore, this study's main contribution is to bring together a broad range of scattered factors across numerous publications. On the other hand, a mixed approach, including survey data from 224 senior and middle managers working in Iranian

SMEs, as a data collection tool, and SEM methodology for the data analysis has not been utilized in the context of BDA.

In this study, besides developing a comprehensive model to describe how BD components affect firms' performance, it is helped to formulate a theoretical basis to create a better understanding of the BDA process for both practitioners and academics. Therefore, this study aims to demonstrate the importance of adopting BD in SMEs as a strategic requirement in Iran by addressing the following research questions.

1. How do the various components of BD affect SMEs' performance?
2. What are the main factors influencing the BDA among SMEs?

To achieve this goal, the remainder of this article is structured as follows. The second section provides a brief overview of the theoretical basis and related works in the literature to achieve this goal. The third section describes the research method, statistical population, and the samples under study. In the fourth section, the research findings are presented. Finally, in the last section, the conclusion and suggestions for further research are presented.

## 2. Literature Review

Given that the effect of BDA on SMEs' performance is the focus of this study, related studies will be reviewed in this section.

A systematic review of the literature on SME digitalization was performed by Meier [20]. The researcher analyzed 77 articles from a broad range of business and management literature. The results confirm that SMEs are composed of firms with a broad range of heterogeneity in the way they approach the challenges of digitalization. It is necessary to discover and analyze the essential determinants affecting the adoption of BIS in healthcare SMEs. So, in another systematic literature review [21], researchers investigated the adoption of business intelligence systems in small and medium enterprises in the healthcare sector. The results of the analysis indicated 15 determinants were significant, and a theoretical structure was developed based on technology, organization, environment, and CEOs' determinants and theories.

Park et al. [22] identified and prioritized the factors impacting BDA using the technology–organization–environment (TOE) framework and the analytic hierarchy process (AHP) method.

Sun et al. [23], by applying an integration of diffusion of innovation theory (DOI), institutional theory, TOE, and content analysis, identified 26 factors affecting BDA in firms and developed a framework to help decision making about BDA in organizations.

Sen, Ozturk, and Vayvay [14] provided a base for future BD research for SMEs by investigating critical potentials and threats that need to be addressed. Additionally, they suggested potential practices to seize the full potential of BD in SMEs.

Another study revealed that infrastructure is the most critical factor for BDA in SMEs. Additionally, possible problems for implementing BD in SMEs were discussed [24].

Brynjolfsson and Hitt [25] studied the adoption rate of BD in US SMEs, and their findings show that SMEs are very interested in using business analytics. They examined the relationship between data-driven decision making and firms' performance.

Tian, Hassan, and Razak [26] reported that BD could be a source of economic strength in the future, as the development of BD has changed China's financial markets by creating a variety of financial and monetary services and tools. They predicted that existing structures would undergo serious changes.

Al Tawara and Gide [27] investigated the effects of mobile technology (MT) and BD-based social media marketing (SMM) adoption in Jordanian SMEs. They conducted a review of MT practices using SMM in SME organizations to determine various variables that could impact the success of BD-based social media platforms adoption in SMEs.

Another study examined the importance of BD in the "data-driven innovation orientation" of organizations. The results showed that the knowledge generated from BD processes improves organizational performance and enables organizational decision makers to resolve issues innovatively [1].

Following a hybrid approach using data collection with a survey of managers and three case studies, Mikalef, Boura, Lekakos, and Krogstie [28] studied various resources and contextual factors that drive performance increases due to investing in BD analytics. Finally, by applying a "fuzzy-set qualitative comparative analysis" method, four different patterns of components in BD analytic resources were identified that lead to high performance.

The impact of BD analytics on Romanian companies' supply chain performance was studied by Oncioiu et al. [29]. Additionally, there have been some challenges, such as assessing tools and technologies such as cloud computing and security technologies, that companies, to achieve proper performance within the supply chain, intend to implement.

Maroufkhani et al. [11] provided a systematic review of BD analytics studies to identify various factors that influence BDA on organizations and determine different performance types that BD may improve.

Mbassegue, Escandon-Quintanilla, and Gardoni [30] discussed contributions related to BD tools and knowledge management in SMEs. They investigated how BD could enable SMEs to improve their competitiveness and performance and their challenges when adopting these technologies.

In another study, the researchers identified barriers preventing SMEs from the successful adoption of BD and proposed a maturity model for SMEs as a roadmap to structure data analytics [31].

Table 1 shows the summary of studies that investigated the impact of BDA in firms' outcomes. The literature shows that despite the growing number of studies on BDA and firm performance, there are lacking attempts to evaluate the impact of BDA on SMEs' performance and examine required conditions to improve firm outcomes through BDA. Therefore, the present study sought to develop existing knowledge in this field.

**Table 1.** Summary of related studies on BDA and firms' performance.

| Reference | Context | Method | Purpose |
|---|---|---|---|
| [32] | Brazilian banks | Interviews | • To reveal the significant role of managers in successful BDA. |
| [22] | Korean firms | AHP, TOE | • To identify and prioritize the factors influencing BDA. |
| [23] | Related papers | DOI, institutional theory, and TOE | • To identify a set of factors affecting BDA in firms. |
| [18] | North American companies | Survey data | • To provide a resource-based view to understand the role of firm critical resources in the context of BDA on firm performance. |
| [19] | Spanish companies | QCA | • To analyze the role of utilizing BD analytics in the firms' innovation process. |
| [33] | Colombian businesses | QCA | • To describe the relation between data maturity and business performance in the context of BDA. |
| [8] | Australian state emergency service | Systematic literature review | • To provide a general view of BD and its role in gaining business value. |
| [34] | The Dutch Tax organization | Interviews | • To describe how BDA can improve decision making quality. |
| [25] | Large firms in the US | Survey data | • To examine the relationship between data-driven decision making and firms' performance. |
| [27] | Jordanian SMEs | Semi-structure theme Interviews | • To investigate the adoption of MT and use of BD-based SMM. |
| [28] | Greek firms | Survey data, QCA | • To investigate factors/resources that cause performance increases from BD analytics investments. |

| Reference | Context | Method | Purpose |
|---|---|---|---|
| [29] | Romanian supply chain companies | Sampling survey, using a questionnaire | • To analyze the impact of BD analytics on supply chains.<br>• To assess the tools needed to achieve proper performance. |
| [11] | Related papers | Systematic literature review | • To provide a systematic review of BD analytics and firm performance.<br>• To identify the factors that may impact the adoption of BD analytics. |

## 3. Research Methodology

### 3.1. Research Design

In an attempt to create a conceptual research model (Figure 1), the related literature, particularly research conducted by [1,11,14,28,29,35–37], were studied. Finally, according to our previous work [6], it was found that 51 factors in 12 groups affect the BDA in organizations. Table 2 provides definitions of the constructs used in the model. Therefore, in the proposed model, first, the impact of this group of factors on BDA is tested. Then, the BDA effect on organizational performance is investigated. In this model, BD components are independent variables, and organizational performance variables, including economic performance (ECP), operational performance (OPP), social performance (SOP), and organizational performance (ORP), are dependent variables. However, the organizational performance variable is defined as the final dependent variable in the model.

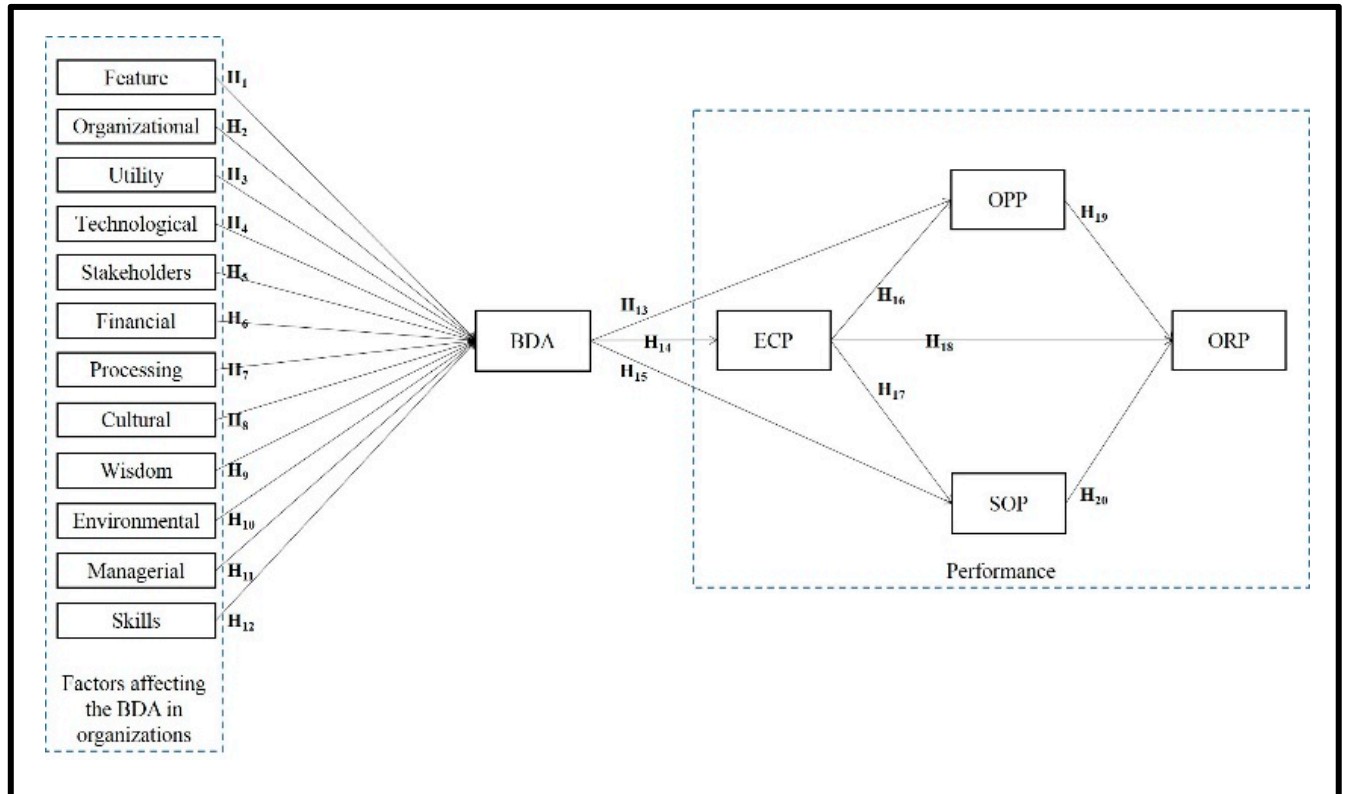

**Figure 1.** Conceptual model.

**Table 2.** Constructs description.

| Construct | Definition |
| --- | --- |
| Feature factors (FEF) | Including: Trainability, Perceived simplicity, Observability, Complexity, and Data quality and integration. |
| Organizational factors (ORF) | Including: Organizational data environment, Business strategy orientation, Firm size, and Industry type. |
| Utility factors (UTF) | Including: Perceived compatibility, Appropriateness, Perceived benefits (advantage), Relative advantage, and Perceived usefulness. |
| Technological factors (TEF) | Including: Technology readiness/technology resources, Wireless technology, Availability of BD tools, Internal versus external technologies, IS competence/IT structure (infrastructure), Technological capability, and Network challenges. |
| Stakeholder factors (STF) | Including: Vendor support, Competitive (Perceived industrial) pressure, Government support, laws and policy, Trading partner adoption/readiness, and IS fashion. |
| Financial factors (FIF) | Including: Perceived financial readiness, Perceived cost, and Cost of adoption. |
| Processing factors (PRF) | Including: system integration, Security and privacy, Data control, Predictive analytics accuracy, and Interpret unstructured data. |
| Cultural factors (CUF) | Including: Organizational (learning) culture, information security culture, and Decision-making culture. |
| Wisdom factors (WIF) | Including: IT expertise, Knowledge about BD, and BD awareness. |
| Environmental factors (ENF) | Including: Supply chain connectivity, Risks of outsourcing, Market turbulence, and Marketing and inventory. |
| Managerial factors (MAF) | Including: Leaders' attitude towards change, Management support for BD, Change efficacy, and Willingness to change. |
| Skill factors (SKF) | Including: Human resources, Staffing, and Training. |
| Economic performance (ECP) | Economic performance reflects the firm's ability to reduce costs associated with purchasing materials, energy consumption, waste management, environmental fines, etc. [38]. |
| Operational performance (OPP) | Operational performance (OPP) is related to the firm's capability to increase product distribution efficiency to customers [39]. |
| Social performance (SOP) | Social performance (SP) reflects the status of organizational beliefs about social responsibilities, social accountability methods, policies, plans, and the evident outcomes associated with the organization's social relationships [40]. |
| Organizational performance (ORP) | Financial and marketing and performance of the organization compared to the industry average [41]. |

*3.2. The Role of SMEs in IRAN*

The main factors determining SMEs are "employee number", "turnover", or "balance sheet total", presented in Table 3. There is no unanimity about a unique definition of SMEs in Iran. Based on the "Ministry of Industries and Mines" definition, enterprises run by 100 workers or less are considered SMEs. About 95% of Iranian manufacturing companies are SMEs. It is estimated that enterprises with less than nine employees constitute about 96.1% of all Iran enterprises. Firms with 10–49 employees are about 3.3%, firms with 50–99 employees 0.3%, and firms with more than 100 employees are 0.4%, respectively [42].

**Table 3.** SMEs' key factors.

| Category | Employees | Turnover | Balance Sheet Total |
| --- | --- | --- | --- |
| Micro enterprises | 1–9 | ≤€ 2 m | ≤€ 2 m |
| Small enterprises | 10–49 | ≤€ 10 m | ≤€ 10 m |
| Medium-sized enterprises | 50–249 | ≤€ 50 m | ≤€ 43 m |

According to Antoldi, Cerrato, and Depperu [43], SMEs have a more significant market share in developing countries than in developed countries. SMEs play a critical role in developing countries' economic growth, especially in employment generation, poverty reduction, and the establishment of a competitive environment. Sadeghi [42] reported, "Iran is the second-largest economy in the Middle East North Africa region in terms of GDP." Therefore, BDA in SMEs can bring considerable benefits to Iran and be the critical factor for driving economic development.

*3.3. Hypotheses*

Based on the literature review and conceptual model shown in Figure 1, this study aims to test 20 hypotheses.

**Hypothesis 1 (H1).** *Feature factors have a positive and direct impact on BDA.*

**Hypothesis 2 (H2).** *Organizational factors have a positive and direct impact on BDA.*

**Hypothesis 3 (H3).** *Utility factors have a positive and direct impact on BDA.*

**Hypothesis 4 (H4).** *Technological factors have a positive and direct impact on BDA.*

**Hypothesis 5 (H5).** *Stakeholder factors have a positive and direct impact on BDA.*

**Hypothesis 6 (H6).** *Financial factors have a negative and reverse impact on BDA.*

**Hypothesis 7 (H7).** *Processing factors have a positive and direct impact on BDA.*

**Hypothesis 8 (H8).** *Cultural factors have a positive and direct impact on BDA.*

**Hypothesis 9 (H9).** *Wisdom factors have a positive and direct impact on BDA.*

**Hypothesis 10 (H10).** *Environmental factors have a positive and direct impact on BDA.*

**Hypothesis 11 (H11).** *Managerial factors have a positive and direct impact on BDA.*

**Hypothesis 12 (H12).** *Skill factors have a positive and direct impact on BDA.*

**Hypothesis 13 (H13).** *BDA has a positive and direct impact on operational performance.*

**Hypothesis 14 (H14).** *BDA has a positive and direct impact on environmental performance.*

**Hypothesis 15 (H15).** *BDA has a positive and direct impact on social performance.*

**Hypothesis 16 (H16).** *Environmental performance has a positive and direct impact on operational performance.*

**Hypothesis 17 (H17).** *Environmental performance has a positive and direct impact on social performance.*

**Hypothesis 18 (H18).** *Environmental performance has a positive and direct impact on organizational performance.*

**Hypothesis 19 (H19).** *Operational performance has a positive and direct impact on organizational performance.*

**Hypothesis 20 (H20).** *Social performance has a positive and direct impact on organizational performance.*

Evaluation items and validation of components affecting BDA and performance in the questionnaire were extracted from various studies.

All items were rated on a 7-point Likert scale from 1 (completely disagree) to 7 (completely agree). In this study, field data drawn from an electronic questionnaire focusing on SMEs in Iran are used. Therefore, this study's population consists of all senior managers and middle–level managers working in SMEs in Iran, who answered the questionnaires via email. Using the information available in the existing social network databases about experts and specialized such as industrial and scientific associations, IREXPERT, LinkedIn, etc., and using a judgmental or purposive sampling process for identifying proper experts, a total of 2863 emails were sent in April 2020. In one month, 836 questionnaires were received, and among them, 612 were removed due to incompleteness. Finally, 224 senior and middle managers completed the questionnaire, indicating that the response rate is 7.8%.

## 4. Results

Table 4 represents the demographic characteristics of the samples.

**Table 4.** Sample demographics.

| Characteristic | | N | % |
|---|---|---|---|
| **Gender** | Male | 187 | 83.5 |
| | Female | 37 | 16.5 |
| **Age** | Less than 30 | 33 | 14.7 |
| | 31–40 | 64 | 28.6 |
| | 41–50 | 88 | 39.3 |
| | More than 50 | 39 | 17.4 |
| **Work experience** | Less than 10 | 54 | 24.1 |
| | 11–20 | 78 | 34.8 |
| | 21–30 | 69 | 30.1 |
| | More than 30 | 23 | 10.3 |
| **Job title** | Chief executive officer | 19 | 8.5 |
| | Chief operating officer | 31 | 13.8 |
| | Factory manager | 41 | 18.3 |
| | Marketing and sales manager | 24 | 10.7 |
| | Purchase manager | 30 | 13.4 |
| | Product manager | 10 | 4.5 |
| | Logistic manager | 18 | 8.0 |
| | Supply chain manager | 25 | 11.2 |
| | Chief information officer | 26 | 11.6 |
| **Total** | | **224** | **100** |

Table 5 shows the different industries in which respondents are working. Respondents were from 20 various sectors.

Prior to the hypothesis test, the Kolmogorov-Smirnov (K-S) test was performed to evaluate the data's normality at the significance level of 5%. As indicated in Table 7, the significance level for all structures is greater than 5%. Thus, all data are normally distributed, and it is possible to use the structural equation modeling for the hypotheses test. Furthermore, Table 6 shows the mean and standard deviation of each variable and Cronbach's alpha value for measuring the validity of the questionnaire.

**Table 5.** Sample industry.

| Description | | N | Percentage |
|---|---|---|---|
| Food and kindred products | | 27 | 12.1 |
| Food and kindred products | | 16 | 7.1 |
| Water, sewer, pipeline, and power line | | 3 | 1.3 |
| Electrical equipment and component manufacturing | | 6 | 2.7 |
| Detergent, cosmetic and hygienic products | | 22 | 9.8 |
| Sanitary ware | | 5 | 2.2 |
| Automotive parts manufacturing | | 16 | 7.1 |
| Medical equipment manufacturing | | 7 | 3.1 |
| Machine-made carpets manufacturing | | 3 | 1.3 |
| Textile mills | | 3 | 1.3 |
| Tobacco products manufacturing | | 4 | 1.8 |
| Lumber and wood products manufacturing | | 7 | 3.1 |
| Office furniture manufacturing | | 10 | 4.5 |
| Papers and allied products manufacturing | | 6 | 2.7 |
| Chemicals and allied products manufacturing | | 17 | 7.6 |
| Pharmaceutical preparations manufacturing | | 12 | 5.4 |
| Plastics, rubber, and synth resins manufacturing | | 21 | 9.4 |
| Concrete, gypsum, and plaster products manufacturing | | 10 | 4.5 |
| Leather and leather products manufacturing | | 12 | 5.4 |
| Transportation equipment manufacturing | | 17 | 7.6 |
| Ownership | Private | 138 | 61.7 |
| | Government and Semi-Government | 86 | 38.4 |
| | Total | 224 | 100 |

**Table 6.** Normality (K-S) test, descriptive statistics, and validity test.

| Construct | N | K-S | Sig. | Number of Items | Mean | Standard Deviation | $\alpha$ | Composite Reliability (CR) | Average Variance Extracted (AVE) |
|---|---|---|---|---|---|---|---|---|---|
| Feature factors | 224 | 2.107 | 0.358 | 5 | 3.084 | 1.093 | 0.755 | 0.849 | 0.693 |
| Organizational factors | 224 | 1.982 | 0.382 | 4 | 3.407 | 1.002 | 0.983 | 0.760 | 0.601 |
| Utility factors | 224 | 1.838 | 0.083 | 5 | 3.826 | 1.078 | 0.907 | 0.852 | 0.527 |
| Technological factors | 224 | 2.005 | 0.237 | 7 | 3.152 | 0.933 | 0.834 | 0.780 | 0.592 |
| Stakeholder factors | 224 | 2.409 | 0.192 | 5 | 3.333 | 0.980 | 0.927 | 0.822 | 0.679 |
| Financial factors | 224 | 2.080 | 0.261 | 3 | 3.457 | 0.964 | 0.947 | 0.795 | 0.685 |
| Processing factors | 224 | 1.567 | 0.211 | 5 | 3.517 | 1.239 | 0.944 | 0.860 | 0.773 |
| Cultural factors | 224 | 2.333 | 0.307 | 3 | 3.621 | 1.005 | 0.941 | 0.796 | 0.639 |
| Wisdom factors | 224 | 1.585 | 0.370 | 3 | 3.784 | 0.927 | 0.979 | 0.853 | 0.825 |
| Environmental factors | 224 | 1.699 | 0.426 | 4 | 3.419 | 1.141 | 0.818 | 0.922 | 0.698 |
| Managerial factors | 224 | 2.144 | 0.255 | 4 | 3.678 | 1.107 | 0.906 | 0.741 | 0.705 |
| Skill factors | 224 | 2.180 | 0.291 | 3 | 3.606 | 1.159 | 0.829 | 0.931 | 0.820 |
| Economic performance (ECP) | 224 | 1.600 | 0.435 | 6 | 3.114 | 0.954 | 0.753 | 0.819 | 0.541 |

**Table 6.** *Cont.*

| Construct | N | K-S | Sig. | Number of Items | Mean | Standard Deviation | α | Composite Reliability (CR) | Average Variance Extracted (AVE) |
|---|---|---|---|---|---|---|---|---|---|
| Operational performance (OPP) | 224 | 2.371 | 0.283 | 5 | 3.829 | 1.208 | 0.758 | 0.814 | 0.674 |
| Social performance (SOP) | 224 | 1.953 | 0.253 | 5 | 3.458 | 1.095 | 0.883 | 0.725 | 0.766 |
| Organizational performance (ORP) | 224 | 1.836 | 0.118 | 7 | 3.804 | 1.078 | 0.877 | 0.793 | 0.575 |

Confirmatory factor analysis (CFA) was performed to validate the factors affecting BDA and performance. According to Du, Yalcinkaya, and Bstieler [44], elliptical reweighted least squares (ERLS) estimation is used in this study because it provides an unbiased estimate of normal and non-normal data. The ERLS chi-square is 234.84. Other fit statistics are shown in Table 7 that all indicate proper fit.

**Table 7.** Model Fit Statistics.

| Criterion | $\chi^2$ | RMSEA | GFI | AGFI | NFI | NNFI | CFI |
|---|---|---|---|---|---|---|---|
| Result | 234.84 | 0.041 | 0.961 | 0.957 | 0.973 | 0.935 | 0.942 |

Because of the large number of constructs and the small sample size, the proposed conceptual model was evaluated using the structural equation modeling method. LISREL 9.30 software was used to complete both the confirmatory factor analysis required to appraise the measurement model and the structural analysis needed to assess the structural model due to the significant model fit information existing.

Table 7 shows a summary of descriptive statistics. All the correlation coefficients shown in Table 8 are significant at the 1% significance level.

**Table 8.** Correlation coefficient matrix.

| | FEF | ORF | UTF | TEF | STF | FIF | PRF | CUF | WIF | ENF | MAF | SKF | ECP | OPP | SOP | ORP |
|---|---|---|---|---|---|---|---|---|---|---|---|---|---|---|---|---|
| FEF | 0.882 | | | | | | | | | | | | | | | |
| ORF | 0.380 | 0.829 | | | | | | | | | | | | | | |
| UTF | 0.291 | 0.612 | 0.812 | | | | | | | | | | | | | |
| TEF | 0.273 | 0.391 | 0.399 | 0.880 | | | | | | | | | | | | |
| STF | 0.471 | 0.412 | 0.482 | 0.726 | 0.840 | | | | | | | | | | | |
| FIF | 0.516 | 0.614 | 0.464 | 0.647 | 0.631 | 0.823 | | | | | | | | | | |
| PRF | 0.440 | 0.454 | 0.690 | 0.642 | 0.584 | 0.439 | 0.781 | | | | | | | | | |
| CUF | 0.500 | 0.712 | 0.578 | 0.471 | 0.489 | 0.575 | 0.347 | 0.796 | | | | | | | | |
| WIF | 0.451 | 0.546 | 0.559 | 0.635 | 0.531 | 0.567 | 0.353 | 0.561 | 0.814 | | | | | | | |
| ENF | 0.542 | 0.318 | 0.632 | 0.598 | 0.402 | 0.346 | 0.317 | 0.434 | 0.555 | 0.826 | | | | | | |
| MAF | 0.380 | 0.496 | 0.720 | 0.634 | 0.432 | 0.343 | 0.509 | 0.678 | 0.315 | 0.420 | 0.790 | | | | | |
| SKF | 0.428 | 0.631 | 0.694 | 0.450 | 0.328 | 0.515 | 0.458 | 0.436 | 0.612 | 0.575 | 0.490 | 0.874 | | | | |
| ECP | 0.628 | 0.701 | 0.590 | 0.506 | 0.412 | 0.469 | 0.585 | 0.502 | 0.477 | 0.501 | 0.619 | 0.433 | 0.819 | | | |
| OPP | 0.227 | 0.563 | 0.448 | 0.409 | 0.641 | 0.457 | 0.632 | 0.559 | 0.429 | 0.463 | 0.559 | 0.340 | 0.332 | 0.834 | | |
| SOP | 0.589 | 0.538 | 0.527 | 0.485 | 0.418 | 0.308 | 0.596 | 0.429 | 0.306 | 0.403 | 0.367 | 0.452 | 0.570 | 0.497 | 0.803 | |
| ORP | 0.519 | 0.607 | 0.376 | 0.373 | 0.531 | 0.475 | 0.337 | 0.399 | 0.320 | 0.480 | 0.424 | 0.510 | 0.4000 | 0.362 | 0.312 | 0.822 |

Table 9 shows the results of the hypotheses test. The research model was approximately accepted, and only two hypotheses were rejected. The hypotheses H1 to 12 were

accepted in the first set, which indicates the impact of all identified factors on BDA. Acceptance of H13 and H14 reflects the influence of BDA on improving the organization's operational and economic performance. However, by rejecting H15, the impact of BDA on improving social performance cannot be verified. Approval of H16 shows the direct effect of ECP on operational performance. The study's findings do not approve of the positive and significant impact of ECP on SOP, and for this reason, H17 is rejected. Additionally, ECP has a positive and significant impact on ORP, and thus H18 is verified. Finally, H19 and H20, which indicated a direct and meaningful relationship of OPP and SOP with ORP, were also accepted as expected.

**Table 9.** Structural model results.

| Research Hypothesis | | | | Beta Coefficients | Status | Research Hypothesis | | | | Beta Coefficients | Status |
|---|---|---|---|---|---|---|---|---|---|---|---|
| H1 | FEF | → | BDA | 0.27 ** | Supported | H11 | MAF | → | BDA | 0.50 ** | Supported |
| H2 | ORF | → | BDA | 0.39 ** | Supported | H12 | SKF | → | BDA | 0.31 ** | Supported |
| H3 | UTF | → | BDA | 0.37 ** | Supported | H13 | BDA | → | OPP | 0.22 ** | Supported |
| H4 | TEF | → | BDA | 0.29 ** | Supported | H14 | BDA | → | ECP | 0.27 ** | Supported |
| H5 | STF | → | BDA | 0.54 ** | Supported | H15 | BDA | → | SOP | 0.09 | Not Supported |
| H6 | FIF | → | BDA | 0.49 ** | Supported | H16 | ECP | → | OPP | 0.28 ** | Supported |
| H7 | PRF | → | BDA | 0.47 ** | Supported | H17 | ECP | → | SOP | 0.13 | Not Supported |
| H8 | CUF | → | BDA | 0.40 ** | Supported | H18 | ECP | → | ORP | 0.43 ** | Supported |
| H9 | WIF | → | BDA | 0.38 ** | Supported | H19 | OPP | → | ORP | 0.39 ** | Supported |
| H10 | ENF | → | BDA | 0.52 ** | Supported | H20 | SOP | → | ORP | 0.59 ** | Supported |

**: Significant at the $p < 0.010$ level.

## 5. Conclusions and Discussion

Considering the rapid increase in volume and complexity of data due to the emerging advanced technologies and diversity of markets, paying attention to data acquisition and analysis has become an inevitable organizational approach. This issue is becoming more pertinent for SMEs, especially in developing countries that encounter limited resources and infrastructures. Accordingly, this study aimed to evaluate the impact of BDA on SMEs' performance. The literature was first reviewed to achieve this goal, and the most critical factors affecting BDA from previous studies were identified. In the next step, the impacts of BDA on performance were evaluated by obtaining the required data from experts.

This study yields valuable results by identifying the impacts of BDA on SMEs' organizational performance. In summary, economic performance has a direct and positive impact on operational and organizational performance but has no significant effect on the organization's social performance. Finally, operational and social performance has a positive and significant impact on organizational performance. These results indicate that to improve various organizational performance measures such as ENP, OPP, SOP, and ORP, the senior management has to commit to BDA.

This study highlights the importance of understanding BD's potential benefits for better decision making and performance improvement because using BD creates many strategic and profitable opportunities for SMEs to succeed in a business ecosystem where competitiveness and innovation are key drivers. BD is a new opportunity for SMEs to use new data types to improve organizations' agility, solve complex problems, and achieve better results and performance. As a result, fundamental changes occur in organizations' operations and performance, and consequently, organizations move towards more accurate and better information-based decision making and modeling. Therefore, using BD is a crucial resource for SMEs to create value, new knowledge, and new processes and products.

This paper has at least three theoretical contributions. First, it is found that 51 factors in 12 groups affect the BDA in organizations. Second, it empirically validates the impact of BDA on different aspects of firm performances. Third, it contributes to the stream of literature on BDA, highlighting another positive effect of developing BDA capabilities.

When using big data as a resource [45], we can conclude that competitive advantage can be obtained and retained by creating and integrating new big data-related capabilities.

Some previous studies [1,6,10,11,14,28,29,35,36] have tried to examine the impact of big data adoption on performance, but this theme in SMEs was probably neglected. Additionally, our proposed framework, in comparison with these studies, considers almost all the factors influencing big data adoption, while previous studies have considered a much smaller number of factors. Anwar et al. [36] found that big data technological capabilities and big data personal capabilities have a significant positive impact on firm performance that is consistent with our research. Yadegaridehkordi et al. [10] also supported this idea and asserted that organizational factors meaningfully impact SMEs' performance. Mikalef et al. [28] considered managerial factors in BDA as the sources which affect organizational performance. Previous studies have also found the importance and positive effects of financial factors in the adoption of big data [26,29]. Data safety and privacy mechanisms, as well as data quality, are still top concerns in big data adoption decisions, according to Chen et al. [46]. Even though the advantages of big data adoption can improve the efficiency of organizations and individuals, security and privacy concerns continue to stymie its adoption [47]. As a result, in the adoption of big data in organizations, privacy and security monitoring techniques for data administration and storage should be addressed. Similar to Awiagah et al. [48], environmental factors were considered as another important factor in BDA. Generally, environmental factors are widely recognized as having a major effect on an organization's readiness to become involved in BDA.

The findings of this study provide managers with a practical framework for decision making about BDA. It can also be useful for managers and entrepreneurs, namely by offering them effective strategies and tips they can adopt and prioritize to improve organizational performance. For instance, they can see how the various components of BD affect an organization's performance. Researchers also may use this study to investigate various relevant hypotheses and improve the accuracy of future studies. Future research can develop variable measurement scales to validate the proposed model. As this model is likely one of the first that evaluates the impact of BDA on SMEs' performance comprehensively, re-evaluating the model using data obtained from various samples could be interesting. Future research may also concentrate on other important resources for big data utilization in SMEs, such as the development of a data-driven culture, KM capabilities, or employee technical skills in terms of education and training relevant to big data-specific competencies. In this study, two major limitations could be addressed in future research to obtain more reliable results. In this study, mediator and moderator variables such as financial resources were not investigated, and therefore, other researchers can consider mediator and moderator variables. Additionally, as the study's focus is on Iranian SMEs, variables such as industry type, organization size, and employee number could be considered control variables.

**Author Contributions:** Conceptualization, methodology, data analysis, writing—review and editing, M.N., J.R. and M.S. All authors have read and agreed to the published version of the manuscript.

**Funding:** The authors did not receive any specific funding for this study.

**Institutional Review Board Statement:** Not applicable.

**Informed Consent Statement:** Not applicable.

**Data Availability Statement:** The data presented in this study are available on request from the corresponding author.

**Acknowledgments:** This work was supported by the Portuguese Foundation for Science and Technology (FCT) and the Center of Technology and Systems (CTS).

**Conflicts of Interest:** There are no financial or non-financial competing interests for any of the authors.

**Abbreviations**

| | |
|---|---|
| SME | Small and medium enterprise |
| BD | Big data |
| BDA | Big data adoption |
| IT | Information technology |
| KMC | Knowledge management cycle |
| SMM | Social media marketing |
| ECP | Economic performance |
| OPP | Operational performance |
| SOP | Social performance |
| ORP | Organizational performance |
| FEF | Feature factors |
| ORF | Organizational factors |
| UTF | Utility factors |
| TEF | Technological factors |
| STF | Stakeholder factors |
| FIF | Financial factors |
| PRF | Processing factors |
| CUF | Cultural factors |
| WIF | Wisdom factors |
| ENF | Environmental factors |
| MAF | Managerial factors |
| SKF | Skill factors |
| ECP | Economic performance |
| OPP | Operational performance |
| SOP | Social performance |
| ORP | Organizational performance |
| KM | Knowledge management |

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
