# Peer review of "The Impact of Big Data Adoption on SMEs’ Performance"

_2504-2289, doi:10.3390/bdcc5040068_

Round 1
Reviewer 1 Report
Dear Authors,
Entitled “The Impact of Big Data Adoption on SMEs Performance” is a theoretical study. This study investigates the impact of BDA on SME’s performance by obtaining the required data from experts. However, the literature section must be strengthened, not just list the references but explain the relationship specifically. Please also check the spelling of some words, though on the whole the language use is okay.
Author Response
We appreciate the time and careful analysis made by the reviewer as well as the recommendations that helped us improve our paper.
We have made many corrections based on reviewers’ ideas.

Reviewer 2 Report
Thank you the authors, this is really interesting paper. This study aims to demonstrate the importance of adopting BD in SMEs as a strategic requirement.
This paper has a scientific significate, as the authors state: "this study's main contribution is to bring together a broad range of scattered factors across numerous publications and including survey data from 224 senior and middle managers working in Iranian SMEs, as a data collection tool, and SEM methodology for the data analysis has not been utilized in the context of BDA."
And has a practical-applied sounder "developing a comprehensive model to describe how BD components affect firms' performance" .
- The introductory part of the article is sufficient, the scientific problem is well designed.
However, systematic literature analysis/review/ could be stronger presented and more primary sources included. This part is poor. - This study aims to test 20 hypotheses. It is too much. In my opinion, this is too fragmented. Authors could further structure the design of the study: by naming the research criteria and indicators, which would allow for a more qualitative presentation of the research logic and construction of more structured hypothesis.
- Sampling type: Sampling type should be justified.
- In the first paragraph, we find a repetition of the text...
- About the presenting the Model; reading this in graphic form would have been clearer.
Author Response

(The authors gave the same response as above.)
